# Advancing Cancer Treatment: Enhanced Combination Therapy through Functionalized Porous Nanoparticles

**DOI:** 10.3390/biomedicines12020326

**Published:** 2024-01-31

**Authors:** Kibeom Kim, Myoung-Hwan Park

**Affiliations:** 1Convergence Research Center, Nanobiomaterials Institute, Sahmyook University, Seoul 01795, Republic of Korea; mpark@syu.ac.kr; 2Department of Chemistry and Life Science, Sahmyook University, Seoul 01795, Republic of Korea; 3Department of Convergence Science, Sahmyook University, Seoul 01795, Republic of Korea

**Keywords:** porous nanoparticle, combination therapy, nanomedicine, functionalization

## Abstract

Cancer remains a major global health challenge, necessitating the development of innovative treatment strategies. This review focuses on the functionalization of porous nanoparticles for combination therapy, a promising approach to enhance cancer treatment efficacy while mitigating the limitations associated with conventional methods. Combination therapy, integrating multiple treatment modalities such as chemotherapy, phototherapy, immunotherapy, and others, has emerged as an effective strategy to address the shortcomings of individual treatments. The unique properties of mesoporous silica nanoparticles (MSN) and other porous materials, like nanoparticles coated with mesoporous silica (NP@MS), metal–organic frameworks (MOF), mesoporous platinum nanoparticles (mesoPt), and carbon dots (CDs), are being explored for drug solubility, bioavailability, targeted delivery, and controlled drug release. Recent advancements in the functionalization of mesoporous nanoparticles with ligands, biomaterials, and polymers are reviewed here, highlighting their role in enhancing the efficacy of combination therapy. Various research has demonstrated the effectiveness of these nanoparticles in co-delivering drugs and photosensitizers, achieving targeted delivery, and responding to multiple stimuli for controlled drug release. This review introduces the synthesis and functionalization methods of these porous nanoparticles, along with their applications in combination therapy.

## 1. Introduction

Cancer remains a significant global health issue, and extensive efforts and research have been devoted to understanding cancer and advancing its treatment [1,2]. To date, various treatment methods have been developed, with approaches such as chemotherapy (CHT) [3], photodynamic therapy (PDT) [4,5,6,7], photothermal therapy (PTT) [8,9,10,11,12], immunotherapy (IMT) [13,14,15,16], cancer starvation therapy (CST) [17,18,19,20,21], thermodynamic therapy (TDT) [22,23,24,25,26,27], chemodynamic therapy (CDT) [28,29,30,31,32,33], magnetothermal therapy (MTT) [34,35,36,37,38], and radiotherapy (RT) [39,40,41]. However, each treatment method has its own limitations and disadvantages. For example, CHT-induced multidrug resistance (MDR), which is one of the major causes of cancer treatment failure. CDT has the challenge of the limited H_2_O_2_ content in the tumor microenvironment (TME). PDT has the challenge of the low photosensitizer stability and high dependence on O_2_. PTT has limitations in treating cancer in deep tissues because light has difficulty penetrating deeply into tissues. The immune system activated by IMT can attack normal tissues, causing immune-related complications [42,43,44]. Thus, the development of new treatment methods is necessary to effectively treat cancer patients [5,45,46].

Recently, there has been growing interest in combining two or more different treatment methods, known as combination therapy, to effectively treat cancer while addressing the shortcomings of individual treatments [43,47,48,49,50]. CHT and IMT, when combined, can reduce the side effects and enhance the immune response to tumor cells, proving effective in treating various types of cancers [51,52,53,54]. The combination of CHT and PTT can enhance drug absorption and effectiveness by generating heat during phototherapy, improving CHT drug delivery to the tumor site and enhancing the cytotoxic effects [55,56,57,58,59]. Moreover, PTT induces the release of tumor antigens and activate immune cells. Therefore, when combined with IMT, PTT can further stimulate immune responses, resulting in improved treatment outcomes [60,61,62,63]. The combination of PDT and CHT allows for effective cancer treatment with smaller drug doses, as PDT generates reactive oxygen species (ROS) that sensitize the TME to the effects of CHT [64,65,66]. Moreover, it can target cancer cells through different mechanisms, making this combination therapy effective against a broader range of cancer types and reducing the risk of drug resistance. In addition, various therapeutic methods are combined and have been used in cancer treatment research [67,68,69,70,71].

However, combination therapy still faces challenges related to drug solubility, biostability, non-specific distribution, side effects, and tumor penetration [72,73,74,75,76,77,78]. Therefore, research efforts have also been directed toward overcoming these issues by utilizing mesoporous nanoparticles [79,80,81,82,83,84]. Mesoporous nanoparticles, with their high surface area and porosity, can effectively encapsulate and deliver various therapeutic agents, addressing the limitations associated with traditional treatments [85,86,87,88,89]. Additionally, the post-functionalization of mesoporous nanoparticles with ligands, biomaterials, and polymers plays a crucial role in unlocking the full potential of these materials for combination therapy [90,91,92,93]. This review paper introduces various mesoporous nanoparticles used in combination therapy, the substances used for post-functionalization, and the post-functionalization methods used in the past five years.

## 2. Porous Nanoparticles for Combination Therapy

### 2.1. Mesoporous Silica Nanoparticles

Mesoporous silica nanoparticles (MSNs) are classified by the FDA as ‘Generally Recognized as Safe’ (GRAS) and are used in cosmetics and as food additives [94]. An MSN, known for its high biocompatibility, is a porous material widely used for drug delivery [95,96,97]. The particles can be adjusted by size and porosity, and they have a high surface area. Drugs can be loaded into the pores of MSNs, and the adjustable size and chemical properties of these pores allow for controlled drug release [98,99,100]. Additionally, the high surface area facilitates functionalization, enabling cancer-targeting ligand modification for targeted cancer therapy [101,102].

Post-functionalization is led by changing the functional group on the surface of MSNs. A compound containing silane is used to induce a siloxy bond with MSNs, and the functional group at the other end is used for functionalization. Yan et al. developed a pH-responsive and cancer-targeting system combining chemotherapy and phototherapy by loading doxorubicin (Dox) and pheophorbide a (PA) into hollow mesoporous silica nanoparticles (HMSNs), followed by functionalization with chitosan (CS) and folic acid (FA) (Figure 1A) [103]. HMSNs, with their extensive surface area, allow co-encapsulation of the widely used chemotherapy drug, Dox, and the photosensitizer, PA, which facilitate CHT, PDT, and PTT. Functionalized CS can block the pores of HMSNs, inhibiting drug release, and can also induce drug release under acidic conditions via protonation of the amine groups in CS, leading to shell swelling. FA interacts with FA receptors, which are overexpressed in cancer cells, enabling targeted cancer therapy. Before functionalization of HMSNs, CS interacts with glycidoxypropyl-trimethoxy-silane (GPTMS) through an epoxy amine reaction, where the silane in GPTMS forms siloxy bonds with the nanoparticle surface. FA is functionalized by forming amide bonds between the amine groups in CS and the carboxylic acid in FA, after CS has been functionalized onto HMSNs. To investigate cellular uptake of the system (HMSNs-GM-CS-FA@Dox/PA), normal L-02 cells and FA-receptor overexpressing KB cells were examined using a confocal laser microscope (CLSM). After incubation with HMSNs-GM-CS-FA@Dox/PA, weak fluorescence was observed in L-02 cells, while strong fluorescence was observed in KB cells (Figure 1B). This result suggests that FA modified the system and facilitated cancer-targeting delivery. Furthermore, to confirm the combination therapy effect, HMSNs-GM-CS-FA@Dox/PA was injected into tumor-bearing mice. The group treated with HMSNs-GM-CS-FA@Dox/PA without light irradiation showed a greater tumor growth inhibition effect compared to the other groups. However, the most significant inhibition of tumor growth was observed in the group treated with HMSNs-GM-CS-FA@Dox/PA when exposed to light irradiation. This result indicates that co-encapsulating HMSNs-GM-CS-FA with DOX and PA can achieve a better combined treatment effect in inhibiting tumor growth by integrating photothermal, photodynamic, and chemotherapy in a single formulation, thereby enhancing therapy effectiveness.

Dopamine has abundant catechol and amine groups and can form a polydopamine (PDA)-modified layer on almost any surface via self-polymerization. This can be used for functionalization by coating silica nanoparticles with dopamine [104]. Lei et al. developed a system that responds to multiple stimuli by coating MSNs with PDA, enabling controlled drug release of a combination of PTT and CHT (Figure 2A–C) [105]. Moreover, 3-mercaptopropyltrimethoxy silane (MPTMS) was used to introduce thiol groups into MSNs. Then, 3-mercaptopropionic acid was introduced to form the disulfide bonds in MSNs with carboxylic acid groups. After loading Dox as the chemotherapy drug, they coated it with PDA. The PDA coating was prepared through the oxidative self-polymerization of dopamine in a mild alkaline aqueous medium with a pH of 8.5. First, dopamine was oxidized and self-polymerized spontaneously at room temperature, forming 5,6-dihydroxyindole through intermolecular cyclization. Finally, PDA was formed through the polymerization reaction. The PDA coating seals the pores of MSNs to prevent drug leakage and allows PTT in response to near-infrared light. MSN-SS-PDA/Dox, which is loaded with Dox and coated with PDA, responds to stimuli such as pH, glutathione (GSH), and near-infrared light irradiation to release the drug (Figure 2D–G). MSN-SS-PDA/Dox displayed a higher drug release rate at pH 5.0 compared to pH 7.4, which was attributed to the partial peeling of the PDA coating on nanoparticles. Additionally, 3-mercaptopropionic acid modified via the disulfide bond of MSNs is removed by GSH, which also induces the removal of PDA, resulting in drug release. Finally, NIR irradiation induces a temperature increase, further promoting drug release by disrupting the electrostatic interactions between the drug and the carrier. To validate the effects of combining CHT and PTT, MSN-SS-PDA/Dox was injected into 4T1 tumor-bearing mice. The single CHT and PTT treatments showed significant tumor suppression. Meanwhile, in the CHT and PTT combination therapy, effective tumor growth inhibition was observed, which could be attributed to the synergistic effect.

Additionally, MSNs can be used not only for cancer therapy but also for imaging in cancer diagnosis. Huang et al. developed a system that enables both PDT and PTT combination therapy, as well as fluorescence and ultrasound (US) imaging, by loading MSNs with indocyanine green (ICG) and perfluorohexane (PFH) (Figure 3A) [106]. ICG, a photosensitizer that can be activated by 808 nm NIR light, has shown an extraordinary capacity to induce both hyperthermia and ROS for PTT/PDT combination therapy. In addition, ICG has been widely used for medical imaging and diagnostics. However, ICG has complications, like an inherent instability, quick elimination in vivo, and a lack of targeting ability. Due to the acoustic variations between gas and plasma, microbubbles (MBs) are typically used as contrast agents in US imaging, but their relatively large size, blood instability, and lack of targeting capacity hinder their application to cancer diagnosis. Therefore, perfluorohexane (PFH) was loaded into the nanocarrier, which can undergo a ‘liquid-to-gas’ phase transformation in response to laser or high-intensity US. By modifying or loading the two substances into MSNs, these complications can be resolved and effective cancer theranostics can be achieved. After loading liquid PFH into MSNs, PDA is used to decorate the MSN surface and effectively encapsulate the PFH. PDA promotes modifications of ICG, through π-π stacking, and poly ethylene glycol (PEG)-modified folate with thiol groups (FA-PEG-SH), through Michael addition. MSNs loaded with PFH and modified with PDA, ICG, and PEG-FA (MSNs-PFH@PDA-ICG-PEG-FA) enable PDT and PTT combination therapy, as the modified ICG on the surface responds to NIR light irradiation (Figure 3B). Furthermore, the generated heat energy induces a phase transition of PFH, enabling US imaging as well as the breakdown of PDA (Figure 3C). This leads to the release of ICG, preventing ICG’s self-quenching and enabling fluorescence imaging with a strong fluorescent intensity. To verify the imaging effect, MSNs-PFH@PDA-ICG-PEG-FA was injected into tumor-bearing mice. In the MSNs-PFH@PDA-ICG-PEG-FA-treated group, a stronger fluorescence intensity and US signal were obtained at the tumor site compared to the control group. Additionally, it was confirmed that MSNs-PFH@PDA-ICG-PEG-FA effectively suppressed the growth of the tumor. These results suggest that MSNs-PFH@PDA-ICG-PEG-FA facilitates both dual imaging and the combination therapy of PDT and PTT.

Furthermore, other nanoparticles can be functionalized onto MSNs for use in combination therapy. Modified nanoparticles provide new functions for cancer therapy to the system through their unique properties. Ong et al. developed a system capable of combining PTT and IMT by synthesizing extra-large pore mesoporous silica nanoparticles (XL-MSN), then loading them with gold nanoparticles (GNPs) and oligodeoxynucleotides containing unmethylated cytosine phosphorothioate-guanine (CpG) motifs (CpG-ODNs) (Figure 4A) [107]. XL-MSN react with (3-aminopropyl) trimethoxysilane (APTMS) to modify their surface with amine functional groups. The surface of XL-MSN is modified with amine functional groups to carry a positive charge, and the size of the pores is approximately 20–30 nm, allowing 1–2 nm sized negatively charged GNPs to be loaded through the pores via electrostatic interaction. XL-MSNs doped with GNP (Au@XL-MSNs) absorb NIR light to induce a photothermal effect, and thiol-modified Polyethylene Glycol (PEG-SH) and CpG-ODN can be modified through Au–thiol interfacial interactions (Figure 4B). PEG increases the bio-compatibility of the system. CpG-ODNs, a type of pathogen-associated molecular pattern (PAMP), are short synthetic single-stranded DNA molecules containing unmethylated CpG motifs and can be used as an adjuvant in cancer vaccines. Therefore, Au@XL-MSNs modified with PEG and CpG-ODNs (Au@XL-MSN-CpG/PEG) facilitate PTT and IMT combination therapy. To observe the effects of this combination therapy, phosphate-buffered saline (PBS), CpG, Au@XL-MSN/PEG, and Au@XL-MSN-CpG/PEG were injected into tumor-bearing mice. The treatment group with Au@XL-MSN-CpG/PEG and NIR light irradiation showed the highest inhibition of tumor growth compared with the PBS, CpG, or PTT group. This indicates that PTT and IMT combination therapy could effectively treat cancer (Figure 4C).

### 2.2. Mesoporous Silica-Coated Nanoparticles

Mesoporous silica (MS)-coated nanoparticles (NP@MS) are composite nanoparticles with a core–shell structure, where the core is made up of other nanoparticles and the shell is MSNs [108,109]. These MS-coated nanoparticles are synthesized by first creating nanoparticles that consist of material other than silica. Then, tetraethyl orthosilicate (TEOS) is added to the solution containing the nanoparticles. Through the added TEOS sol–gel process, an MS shell is formed on the surface of the nanoparticles. The MS shell has a porous structure, which not only provides space for loading or storing drugs but also provides various functions to the system through easy surface modification [110]. Different nanoparticle possesses have unique physicochemical properties. These properties can facilitate cancer treatment even without the addition of extra therapeutic materials [108,111,112].

Porous polydopamine nanoparticles have received great attention as drug delivery systems due to their high biocompatibility and biodegradability. They are also of great interest for cancer treatment due to their ability to generate heat in response to light. Seth et al. developed mesoporous silica-coated PDA nanoparticles (PDA@MS) for PTT and IMT combination therapy by loading the immunomodulatory drug (gardiquimod (Gardi)) (Figure 5A,B) [113]. The PDA core of PDA@MS possesses photothermal properties. Additionally, an increase in the PDA@MS concentration under NIR light irradiation leads to an increase in the temperature of the solution. The MS shell provides sites for loading Gardi and 1-tetradecanol. Here, 1-tetradecanol, a biocompatible phase-changing material, has a melting temperature of 38–39 °C, slightly higher than normal human body temperature. Therefore, the increased temperature induced by the photothermal properties of PDA causes a phase change of 1-tetradecanol, blocking the pores and triggering the release of the encapsulated drug (Figure 5C,D). Moreover, PTT induces a partial removal of tumors and the release of tumor-associated antigens (TAAs) and damage-associated molecular patterns (DAMPs). The released TAAs and DAMPs, in synergy with Gardi, create a tumor inhibitory environment. To confirm the therapeutic effect of this combination, PBS, PDA@MS, and PDA@MS loaded with Gardi were injected into B16-F10 cancer-bearing mice. It was observed that the group treated with PDA@MSs loaded with Gardi under NIR irradiation had significantly inhibited tumor growth compared to groups treated with PBS and PDA@MS alone. This result indicated that effective PTT and IMT combination therapy was achieved through PDA@MSs loaded with Gardi.

In addition to antibacterial activity, silver nanoparticles (AgNPs) have unique cytotoxic functions against mammalian cells. Therefore, AgNPs are receiving a lot of attention in cancer therapy. In particular, silver ions released from AgNPs cause an imbalance in cellular homeostasis and induce cell death. Zhang et al. developed a system that facilitates the combination therapy of CHT, CST, and ion therapy using mesoporous silica-coated silver nanoparticles (AgNP@MS) (Figure 6A) [114]. They synthesized AgNP@MS modified with glucose oxidase (GOx) (AgNP@MS-GOx) by creating an amide bond between the amine groups of the yolk–shell structured AgNP@MS surface and the carboxylic acid in GOx. Furthermore, the prodrug, tirapazamine (TPZ), was loaded into the silica shell to synthesize the combination therapy system (TPZ-AgNP@MS-GOx). TPZ forms toxic metabolites under hypoxic conditions, damaging DNA and proteins and inducing cell apoptosis. GOx depletes glucose and oxygen, nutrients essential for cancer cells, and promotes the production of gluconic acid and hydrogen peroxide, thus inducing CST. The GOx reaction changes the TME, reducing the oxygen levels and activating TPZ. Additionally, the produced hydrogen peroxide accelerates the oxidation process of Ag, increasing the production of highly toxic Ag^+^ ions in cancer cells. The efficacy of this combination therapy was confirmed by observing the inhibition of cell proliferation after MCF-7 cells were treated with MSN, AgNP@MS, AgNP@MS-GOx, and TPZ-AgNP@MS-GOx. It was observed that TPZ-AgNP@MS-GOx more effectively inhibited cancer cell proliferation compared to other groups (Figure 6B,C).

Gold nanorods (GNRs) not only have high biocompatibility but also generate local heat through the surface plasmon resonance effect caused by light stimulation, facilitating non-invasive PTT in normal cells. Dai et al. developed a system using mesoporous silica-coated gold nanorods (GNR@MS) to facilitate the combination therapy of CHT and IMT (Figure 7A) [115]. They modified the surface of the core–shell structured GNR@MS with 3-triethoxysilylpropylamine (APES) to introduce amine functional groups, and then loaded BMS1166 into the GNR@MS. Afterwards, pegylated anti-vascular endothelial growth factor peptide vaccine (VVP) was modified with BMS1166-loaded GNR@MS through an amide bond to create a combination therapy system based on GNR@MS (GSBVVP). The GNR core in this system generates hyperthermia upon NIR light irradiation, enabling PTT. The MS shell provides a space for encapsulating BMS1166 and allows for post-functionalization with VVP. BMS1166 inhibits the interaction between programmed cell death protein 1 (PD-1) and its ligand, PD-L1, thereby enhancing the effectiveness of IMT. PD-L1, which is overexpressed on cancer cells, interacts with the T-cell receptor PD-1 and blocks T-cell-mediated cancer cell attacks. Therefore, BMS1166 suppresses the cancer cells’ ability to evade T-cell detection through PD-L1. VEGF promotes angiogenesis, and inhibiting it plays a crucial role in treating conditions like hepatocellular carcinoma. Hence, VVP inhibits VEGF, blocking angiogenesis in tumor tissue and enhancing therapeutic efficacy. The hyperthermia generation effect of GSBVVP was observed at various concentrations, with an increase in the temperature of the solution as the concentration increased (Figure 7B). Additionally, to verify the combination therapy effect, GSBVVP was injected into tumor-bearing mice. Under NIR light irradiation, the combination of PTT and IMT provided by GSBVVP showed a more effective tumor growth inhibition rate than each therapy alone. Moreover, it demonstrated the potential to treat metastatic cancer, showing effectiveness toward treating tumors that were not exposed to NIR light.

Wen et al. developed a system utilizing GNR@MS for the combination of PTT and TDT (Figure 8A) [116]. Here, 2,2′-azobis [2-(2-imidazolin-2-yl)propane]-dihydrochloride (AIPH) and lauric acid were loaded onto GNR@MS, followed by modifying the nanoparticle surface with PEG to increase biocompatibility. The GNR core generates heat in response to NIR light irradiation, increasing the local temperature and facilitating PTT. AIPH, a thermally activated alkyl free radical-releasing molecule, decomposes under heat, forming radicals and enabling TDT (Figure 8B). In a hypoxic TME, traditional dynamic therapy has used methods to generate or deliver oxygen and improve treatment efficacy. However, TDT overcomes the limitations of traditional dynamic therapy that depends on the oxygen concentrations by generating radicals through decomposition of the encapsulated material. Lauric acid, a phase-change material (PCM), has a melting point of 43.8 °C, which is higher than normal body temperature. Therefore, lauric acid, in a liquid state, is loaded into the MS and changes into a solid, blocking the pores of the MS and preventing AIPH leakage. The solid lauric acid then changes back to a liquid state due to heat generated from the GNR, opening the pores of the MS and allowing the release of encapsulated AIPH (Figure 8B). Therefore, GNR@MS modified with PEG after loading AIPH and PCM (ASAPP) promotes drug release in response to NIR light and facilitates the combination therapy of PTT and TDT. To confirm the combination therapy effect, PBS, GNR@MS, and ASAPP were injected into tumor-bearing mice. The group treated with ASAPP under NIR light irradiation showed a significantly smaller tumor size compared to groups treated with PBS and GNR@MS (Figure 8C,D). 

### 2.3. Metal–Organic Framework

Metal–organic frameworks (MOFs) are compounds with a three-dimensional structure composed of metal ions and organic ligands [117,118,119]. This structure can be precisely controlled, as MOFs provide a large surface area and volume while retaining stability, authorized by the strong bonds between the metal ions and organic ligands [120,121,122]. The abundant functional groups present on the MOF surface enable interaction with other materials and facilitate the modification of other materials. These properties allow MOFs to be used as drug carriers that control drug release in response to various stimuli or facilitate various cancer therapies [123,124,125,126,127].

PCN-224, a Zr-based porphyrin-based MOF, has high stability even in aqueous solutions and is stable over a wide pH range. In addition, it is attracting much attention as a drug carrier because it can generate ^1^O_2_, which is essential for PDT. Kim et al. developed a system using PCN-224, hyaluronic acid (HA), and Dox that facilitates the combination therapy of CHT and PDT (Figure 9A) [128]. PCN-224 is fabricated through the coordination connection between zirconium and meso-tetra (4-carboxyphenyl)porphine (TCPP). Dox, a well-known anticancer drug, is loaded into the pores of PCN-224 through physical adsorption to synthesize Dox-loaded PCN-224 (Dox-PCN). HA is added to Dox-PCN, where its carboxylic acid forms a coordination bond with zirconium, resulting in an HA-coated Dox-PCN system (HA-Dox-PCN). TCPP, which composes PCN-224, is a porphyrin derivative used as a PDT reagent and can facilitate PDT in response to light irradiation. Additionally, HA, which coats PCN-224, interacts with the overexpressed cd44 receptor in cancer cells, facilitating cancer targeting and blocking the pores of PCN-224 as a gatekeeper. HAdase, which is present in cancer cells, decomposes HA and opens the pores of PCN-224, promoting the release of encapsulated Dox. Thus, HA controls drug release by responding to the enzyme. To confirm the cancer targeting, HA-Dox-PCN was incubated with CD44-negative HEK 293T cells and CD44-positive MDA-MB231 and SCC-7 cells, followed by observation through CLSM. In HEK 293T cells, the fluorescence of Dox was not observed, but a strong fluorescence of Dox was observed in MDA-MB231 and SCC-7 cells (Figure 9B). Furthermore, to confirm the effects of the combination therapy, PCN and HA-Dox-PCN were incubated with HEK 293T and MDA-MB231 cells. In HEK 293T cells, only those treated with light irradiation and PCN showed the inhibition of cell proliferation. In contrast, in MDA-MB cells, both the HA-Dox-PCN-treated group and the HA-Dox-PCN-treated with light irradiation group showed the inhibition of cell proliferation. Notably, the HA-Dox-PCN with light irradiation group exhibited a stronger inhibition of cell proliferation than the HA-Dox-PCN-only group, suggesting this was caused by the combination of CHT and PDT.

MIL-88B(Fe) has carboxylic acid and amino groups that can interact with other substances. Fe^3+^ ions released from MIL-88B(Fe) convert H_2_O_2_ into ·OH through peroxidase-like activity, causing cell death. Zeng et al. developed a system using MOFs with two different phases, ZIF-8 and MIL-88B(Fe), to enable the combination of CHT and CDT (Figure 10A,B) [129]. MIL-88B(Fe) is synthesized through a coordination connection between metal ions (Fe^3+^) and 2-aminoterephthalic acid (NH_2_-BDC). Afterwards, MIL-88B(Fe) is partially etched by adding 2-methylimidazole (2-MeIm) and zinc ions to form hollow MIL-88B(Fe) (hMIL-88B(Fe)). The adsorbed zinc ions on the surface of MIL-88B(Fe) interact with 2-MeIm to form ZIF-8, maintaining the morphology of hMIL-88B(Fe). Dox and manganese oxide nanoparticles (MnO_x_) are loaded into ZIF-8 modified hMIL-88B(Fe) (hMIL-88B(Fe)@ZIF-8), and the surface is modified with folic acid (FA) through a coordination connection. This multifunctional system (hM@ZMDF) is capable of specifically targeting cancer cells and facilitating magnetic imaging, and the combination of CHT and CDT. The decomposition of hMIL-88B(Fe) induces the release of Fe^3+^, which is reduced to Fe^2+^ by overexpressed GSH in cancer cells. This Fe^2+^ then converts H_2_O_2_ into hydroxyl radicals via the Fenton reaction, facilitating CDT. Dox, apart from facilitating CHT, also acts as a fluorophore for fluorescence imaging. MnO_x_ serves as an MRI reagent, allowing for real-time monitoring of drug delivery. FA, known as a cancer-targeting ligand, imparts its cancer-targeting ability to hM@ZMDF through specific interactions with the overexpressed FA receptors on cancer cells. To confirm the cancer-targeting combination effect of hM@ZMDF, both hMIL-88B(Fe)@ZIF-8 and hM@ZMDF were incubated with FA receptor-negative hCMEC/D3 cells, and FA receptor-positive MCF-7 and HepG-2 cells. Incubation of MCF-7 and HepG-2 cells with hMIL-88B(Fe)@ZIF-8 showed cell proliferation inhibition caused by CDT. However, in MCF-7 and HepG-2 cells incubated with hM@ZMDF, a stronger cell proliferation inhibition was observed, attributed to the combined effects of CHT and CDT. Meanwhile, in hCMEC/D3 cells, insignificant cell proliferation inhibition was observed (Figure 10C–H).

MOF-235 is synthesized through a coordination connection between the metal ions (Fe^3+^) and organic linker terephthalic acid (TPA). Similar to MIL-88B(Fe), MOF-235 induces cell death by releasing Fe^3+^ ions. Deng et al. developed a system for the combination of CDT and PTT using MOF-235, PDA, IR820, and piperlongumine (PL) (Figure 11A–C) [130]. Initially, PL is loaded onto MOF-235 to synthesize PL-loaded MOF-235 (MP), and PDA is coated on the surface of MOF-235 through dopamine self-polymerization. By adding IR820 to the PL-loaded and PDA-coated MOF-235 (MP@P), the PDA on the surface interacts with IR820 through π−π stacking and hydrophobic interactions, immobilizing IR820 on the surface of MOF-235, thereby forming a combination system based on MOF-235 (MP@PI). MOF-235 is an effective iron donor, reacting with H_2_O_2_ to generate a significant amount of ROS through the Fenton reaction, facilitating CDT. The Fenton reaction requires a substantial amount of H_2_O_2_, which is provided by PL. PL not only kills cancer cells through the generation of ROS but also increases the level of intracellular H_2_O_2_. PDA possesses excellent biocompatibility and photothermal characteristics. IR820, an NIR dye, generates heat under NIR light irradiation, facilitating PTT in conjunction with the photothermal characteristics of PDA. To confirm the effects of the combination therapy, PBS, PL, MP, MP@P, and MP@PI were injected into tumor-bearing mice. Although the group treated with MP@P showed some tumor growth inhibition by the CDT, the group treated with MP@PI and NIR irradiation exhibited more tumor growth inhibition, facilitated by the combination of CDT and PTT (Figure 11D,E). 

As described previously, the organic ligands that constitute the MOFs can not only exploit their unique properties in cancer therapy but also form coordination compounds with metal ions, providing the system with new functionality. Ding et al. developed a system to facilitate the combination of CHT and CDT using PCN-224 (Fe), GNP, PEG-SH, dodecane thiol (C_12_-SH), and camptothecin (CPT) (Figure 12A–C) [131]. PCN-224(Fe) is synthesized through the coordination connection of zirconium and iron(III) meso-tetra(4-carboxyphenyl)porphine chloride (TCPP(Fe)). When HAuCl4 solution and a reducing agent are added to PCN-224(Fe), gold is reduced on the PCN surface and GNP is formed. The reduced GNP is immobilized on the surface of PCN-224(Fe) to serve as anchors for ligand modification. CPT, which forms π–π stacking with TCPP and a coordination connection with zirconium, is loaded into the pores of GNP-immobilized PCN-224(Fe) (Au/FeMOF). Upon adding PEG-SH and C12-SH to the CPT-loaded Au/FeMOF (Au/FeMOF@CPT), an Au–thiol bond is formed with the surface GNP, synthesizing a system modified with PEG and C_12_ (PEG-Au/FeMOF@CPT). PCN-224(Fe) provides iron, facilitating CDT through the Fenton reaction. GNPs not only serve as anchors for ligand modification but also generate H_2_O_2_ by reacting with glucose, mimicking GOx properties. CPT, a topoisomerase I inhibitor and a well-known cancer drug, facilitates CHT, while the surface-modified PEG and C_12_ ligands enhance the biostability of the system. Particularly, C_12_, with its hydrophobic properties, prevents glucose access until the degradation of the PEG-Au/FeMOF@CPT, triggered by the high concentration of phosphate inside cancer cells (Figure 11A). Thus, iron and CPT are released after cellular internalization, inducing the effects of CHT and CDT. To confirm the effects of the combination therapy, PBS, PEG-Au/FeMOF, CPT, PEG-Au/HMOF@CPT, and PEG-Au/FeMOF@CPT were injected into tumor-bearing mice. The PEG-Au/FeMOF-treated group showed a tumor reduction effect by CDT, while the groups treated with CPT and PEG-Au/HMOF@CPT showed a tumor reduction effect by CHT. However, the most effective tumor reduction effect was observed in the group treated with PEG-Au/FeMOF@CPT, supporting the combination therapy effect of CHT and CDT (Figure 12E).

### 2.4. Other Porous Nanoparticles

Platinum (Pt), known for its high biocompatibility, can be utilized in computed tomography (CT) imaging and possesses the ability to convert laser energy into heat, enabling PTT [132,133,134,135]. Fu et al. developed a system for the combination of CHT and PTT using mesoporous platinum nanoparticles (mesoPt), PEG-SH, and doxorubicin (Dox) [136]. Dox is loaded into the pores of mesoPt, and PEG-SH forms Pt–thiol bonds on the surface of Pt, synthesizing a combination therapy system based on meso Pt (PEG@Pt/Dox). To verify the combination therapy effect, MCF-7/ADR cells were treated with PEG@Pt/Dox and the cell proliferation was monitored. A higher suppression of the cell proliferation rate was observed in those treated with both PEG@Pt/Dox and laser irradiation compared to those treated with only PEG@Pt/Dox.

Carbon dots possess excellent optical properties, including strong fluorescence and good photostability [137,138,139,140]. This facilitates the real-time imaging of cancer cells and tissues, aiding in the diagnosis and monitoring of the treatment response. Additionally, carbon dots generate heat energy by responding to light stimuli, facilitating PTT [141,142,143,144]. Zhang et al. developed a system for the combination of PTT and PDT using porous carbon dots (CDs), triphenylphosphonium (TPP), 5-aminolevulinic acid (ALA), and GNP for CT imaging and targeting mitochondria (Figure 13A) [145]. CDs were synthesized with glycine and citrin through the hydrothermal carbonization method. TPP modifies CDs through amide bond formation between the carboxylic acid group of TPP and the amine group of CDs, forming CDs modified with TPP (T-CDs). When HAuCl4 is added to T-CD, HAuCl4 is reduced on the T-CD surface, forming GNP-doped T-CDs (T-CDs@Au). ALA is loaded into the pores of T-CDs@Au through electrostatic interactions, forming ALA-loaded T-CDs@Au (T-CDs@Au/ALA). TPP imparts its mitochondria-targeting ability to T-CDs@Au/ALA, and the endogenous nonprotein amino acid, ALA, is converted into the photosensitive, protoporphyrin IX, within cells, facilitating PDT. GNP, with a high X-ray attenuation coefficient, can be used as CT contrast agents for tumor imaging. Therefore, T-CDs@Au/ALA targets mitochondria, facilitating the combination therapy of PTT and PDT, as well as providing diagnostic capabilities through CT imaging.

In 2005, Yaghi’s research group first proposed covalent organic frameworks (COFs), which were described as periodic polymer networks constructed via reversible condensation reactions, with a structural potential for biodegradation [146]. COFs possess a sophisticated pore structure and a large surface area, allowing for the loading or storage of drugs [147,148,149]. Their stability and biocompatibility make them a subject of extensive research as drug delivery systems [150,151,152]. Additionally, when COF is synthesized using organic materials used in cancer therapy, combination therapy is possible without additional functionalization. Wang et al. developed a system enabling the combination therapy of PDT and PTT by utilizing COF-366, which is fabricated at the nanoscale (Figure 13B) [153]. COF-366 is a donor-acceptor type of COF that converts light into heat energy and changes triplet oxygen into singlet oxygen, facilitating both PTT and PDT. Additionally, it can be used as a photoacoustic imaging reagent, allowing for simultaneous cancer therapy and therapy monitoring. COF-366 is synthesized via the reaction of tetra (p-amino-phenyl) porphyrin (TAPP) with terephthalaldehyde. To verify the combination therapy effect, COF-366 was injected into tumor-bearing mice for assessment.

**Figure 13 biomedicines-12-00326-f013:**
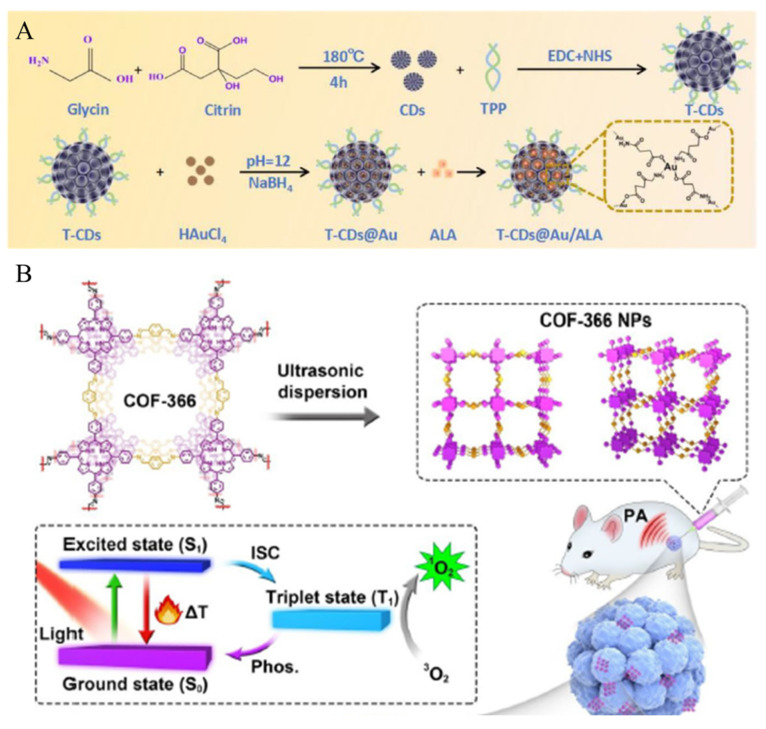
(**A**) Synthesis of T-CDs@Au/ALA. Reproduced with permission from [142]. Copyright (2023): American Chemical Society. (**B**) Schematic illustrations of the formation of COF-366 NPs. Reproduced with permission from [153]. Copyright (2019): Elsevier (Amsterdam, The Netherlands).

Mesoporous polydopamine (MPPD), a melanin-like polymer, has received considerable attention as a drug carrier due to its excellent biocompatibility, high drug loading capacity, and ease of functional modification [154,155,156,157]. Hu et al. developed a system facilitating a combination of PTT and PDT using MPPD, FA-PEG-SH, perfluorooctane (PFO), and IR-820 [158]. FA-PEG-SH is modified on the surface of MPPD through a Michael addition reaction between the thiol group of FA-PEG-SH and the alpha-beta unsaturated carbonyl in MPPD. The combination therapy system (IR-820/PFO@FA-MPPD) is synthesized by loading PFO and IR-820 onto FA-PEG-SH-grafted, MPPD. Structurally similar to ICG, IR820 facilitates both PTT and PDT in response to NIR light, showing improved stability and an improved singlet oxygen quantum yield compared to ICG. Moreover, PFO, known for its high oxygen solubility and used as an oxygen carrier, can deliver oxygen to the hypoxic TME to enhance the PDT effect. To verify the combination therapy effect, IR-820/PFO@FA-MPPD was injected into tumor-bearing mice to compare the tumor growth inhibition rates.

## 3. Conclusions

Various methods utilizing MSNs, NP@MSN, MOFs, mesoPt, CDs, COFs, and MPPD have been explored to facilitate combination therapy. The high biocompatibility, tunable size and surface properties of MSNs make them suitable for a variety of therapeutic strategies, including drug loading and targeted cancer therapy. 

NP@MSN with a yolk–shell or core–shell structure can be made by coating various nanoparticles with MSNs, where the properties of both MSNs and nanoparticles can be utilized for cancer therapy. MOFs composed of metal nodes and organic linkers utilize the properties of both metal and organic linkers in cancer therapy and are especially used in phototherapy and CDT. Other porous nanoparticles, like mesoPt, CDSs, CPFs, and MPPD, have been used in combination therapies, showing an enhanced anticancer effect. Different nanoparticles, enzymes, targeting ligands, organic reagents, photosensitizers, and anticancer agents have been modified using various functionalization methods to further improve the performance of these porous nanoparticles. This kind of modification has endowed systems with multiple functionalities, including biocompatibility, cancer targeting, cancer therapy, and control of drug release. Table 1 summarizes the research on combination therapy using mesoporous nanoparticles. Combination therapies using functionalized porous nanoparticles have made significant advances in the field of cancer therapy, and it is expected that new possibilities for cancer theranostics will be explored through continued research and innovation.

## Figures and Tables

**Figure 1 biomedicines-12-00326-f001:**
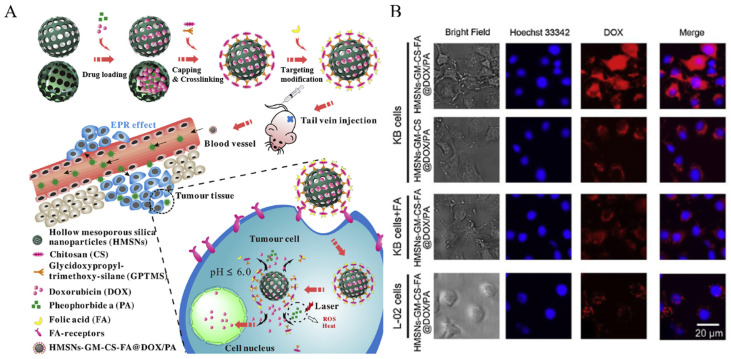
(**A**) The synthesis of HMSNs-GM-CS-FA@DOX/PA and pH-responsive drug release. (**B**) CLSM images of KB and L-02 cells incubated with HMSNs-GM-CS-FA@DOX/PA and HMSNs-GM-CS-FA@DOX/PA with/without excess free FA. Reproduced with permission from [103]. Copyright (2020): Elsevier (Amsterdam, The Netherlands).

**Figure 2 biomedicines-12-00326-f002:**
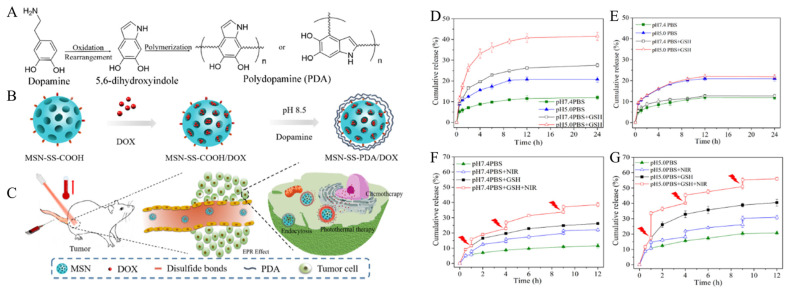
(**A**) Synthesis of PDA film through oxidative polymerization and a Michael addition reaction. (**B**) The synthesis route of MSN-SS-PDA/DOX. (**C**) Combined chemo-photothermal therapy on tumor cells. Cumulative percentage release of (**D**) MSN-SS-PDA/DOX and (**E**) MSN-PDA/DOX in pH 5.0 PBS and pH 7.4 PBS with or without the addition of 10 mM GSH. In vitro drug release profile of MSN-SS-PDA/DOX with NIR laser irradiation in different media (**F**) pH 7.4 PBS and (**G**) pH 5.0 PBS. Red symbol means NIR irradiation. Reproduced with permission from [105]. Copyright (2019): Elsevier (Amsterdam, The Netherlands).

**Figure 3 biomedicines-12-00326-f003:**
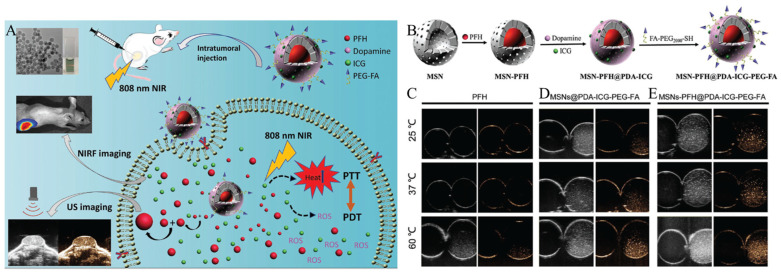
(**A**) Scheme of a dual-model imaging theragnostic system based on mesoporous silica nanoparticles for combined PTT and PDT. (**B**) Fabrication scheme of MSNs-PFH@PDA-ICG-PEG-FA. In vitro US imaging in (**B**) and contrast modes of (**C**) PFH, (**D**) MSNs@PDA-ICG-PEG-FA, and (**E**) MSNs-PFH@PDA-ICG-PEG-FA at different temperatures (25, 37, and 60 °C). Reproduced with permission from [106]. Copyright (2019): John Wiley and Sons (Hoboken, NJ, USA).

**Figure 4 biomedicines-12-00326-f004:**
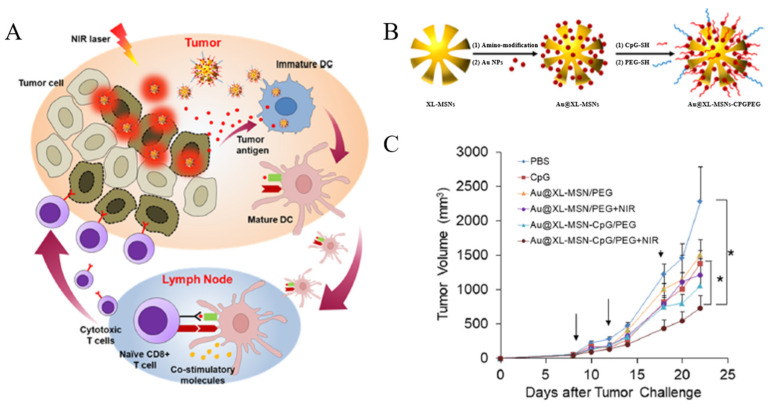
Schematic illustrations of the (**A**) mechanism of Au@XLMSN-CpG/PEG in cancer immunotherapy and photothermal therapy and (**B**) synthesis of PEGylated mesoporous silica nanoparticles decorated with gold nanoparticles for the delivery of thiol-modified CpG-ODNs (Au@XL-MSN-CpG/PEG). (**C**) Tumor volume changes in mice who received PBS, soluble CpG-ODN, Au@XL-MSN/PEG, Au@XL-MSN/PEG with NIR irradiation, Au@XL-MSN-CpG/PEG, and Au@XL-MSN-CpG/PEG with NIR irradiation (* *p* < 0.05). Reproduced with permission from [107]. Copyright (2019): American Chemical Society (Washington, DC, USA).

**Figure 5 biomedicines-12-00326-f005:**
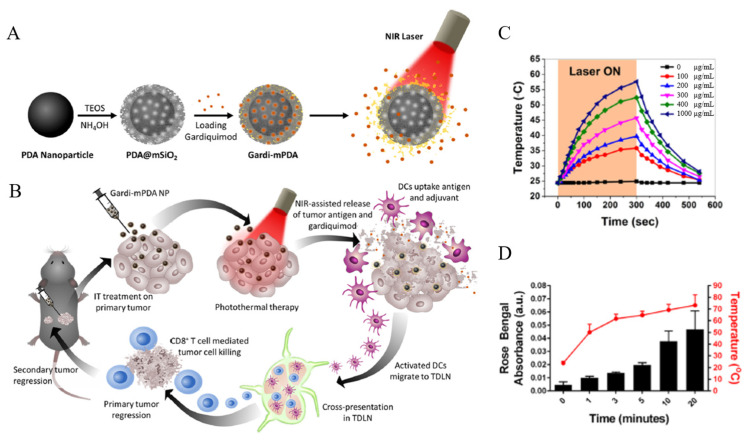
Schematic illustrations depicting (**A**) synthesis of gardiquimod-loaded mesoporous silica-coated polydopamine nanoparticles (gardi-PDA@MS) and NIR-assisted drug release and (**B**) tumor ablation and drug release under NIR irradiation followed by activation of dendritic cells and effector T cells in tumor-draining lymph nodes for regression of primary and secondary tumors. (**C**) Temperature profile and effect of PDA@MS particle concentration on the temperature rise when aqueous solutions were subjected to a laser power density of 14 mW/mm^2^. (**D**) Cumulative release of model dye from the PDA@MS nanoparticles after different laser irradiation durations and their corresponding solution temperature. Reproduced with permission from [113]. Copyright (2020): American Chemical Society (Washington, DC, USA).

**Figure 6 biomedicines-12-00326-f006:**
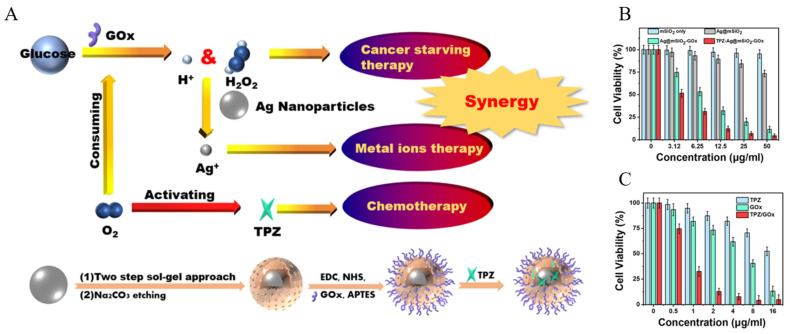
(**A**) Synthesis of the TPZ-Ag@MS-Gox nanoparticles and an illustration of the synergistic combination therapy. (**B**) MCF-7 cell viability rates at different nanoparticle concentrations. (**C**) MCF-7 cell viability rates at different drug concentrations. Reproduced with permission from [114]. Copyright (2020): American Chemical Society (Washington, DC, USA).

**Figure 7 biomedicines-12-00326-f007:**
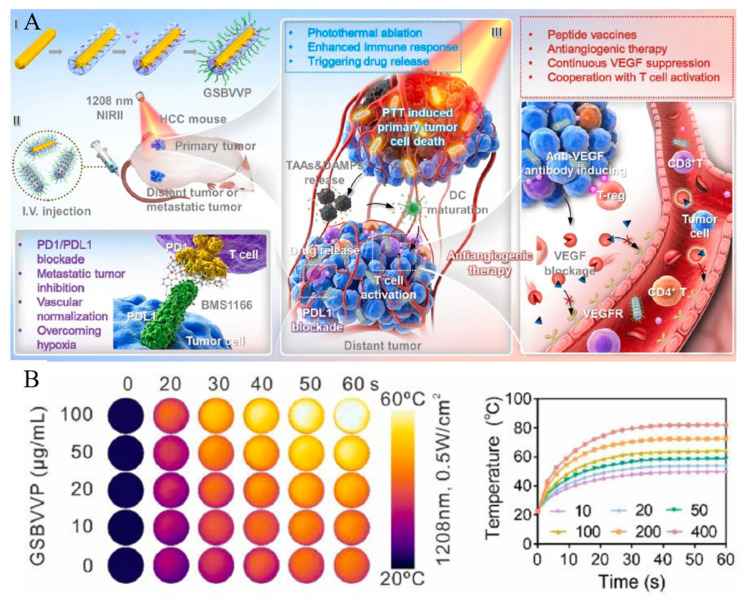
(**A**) Graphical representation of GSBVVP-mediated photothermal immunotherapy. (**B**) Thermal images of different concentrations of GSBVVP solutions under 1208 nm photoirradiation with 0.5 W/cm^2^ exposure intensity for 60 s. Reproduced with permission from [115]. Copyright (2023): Elsevier (Amsterdam, The Netherlands).

**Figure 8 biomedicines-12-00326-f008:**
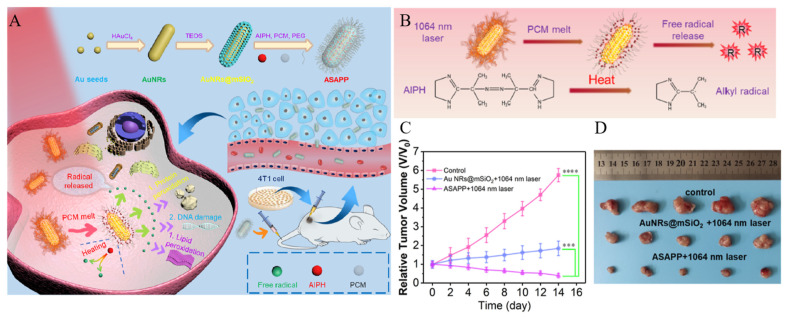
(**A**) Schematic diagram of the NIR-II light-triggered TDT therapy strategy (**B**) Schematic diagram of the NIR-II light-triggered free radical release. (**C**) Tumor growth curve after various treatments (*** *p* < 0.001, **** *p* < 0.0001). (**D**) Digital photographs of the dissected tumors from each group. Reproduced with permission from [116]. Copyright (2023): American Chemical Society (Washington, DC, USA).

**Figure 9 biomedicines-12-00326-f009:**
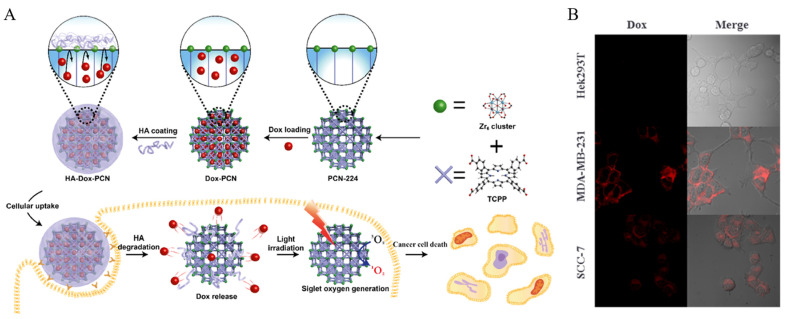
(**A**) Illustration of the HA gatekeeper MOF nanosystem preparation and the combination therapy procedure (**B**) CLSM images to check cellular uptake of Dox-loaded HA-PCN nanoparticles after 2 h of incubation. Reproduced with permission from [128]. Copyright (2019): American Chemical Society (Washington, DC, USA).

**Figure 10 biomedicines-12-00326-f010:**
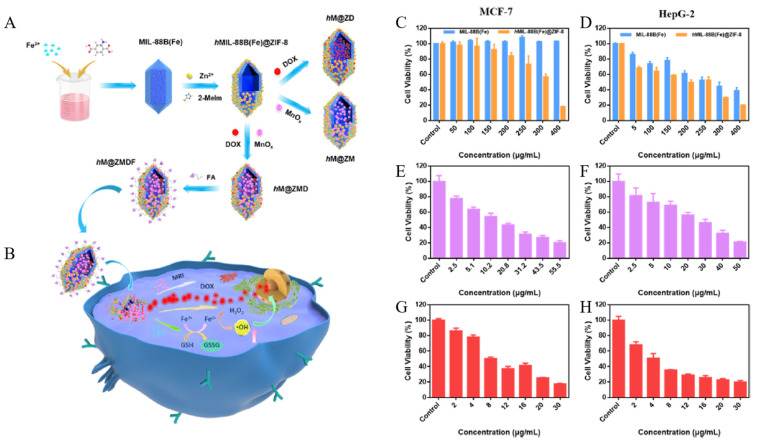
(**A**) Schematic fabrication routes of hM@ZMDF nanoplatforms based on MIL-88B(Fe). (**B**) Schematic illustration of hM@ZMDF nanoplatforms for folate-targeted and combined cancer therapy; nominations: hM@ZM, hMIL-88B(Fe)@ZIF-8-MnOx; hM@ZD, hMIL-88B(Fe)@ZIF-8-DOX; hM@ZMD, hMIL-88B(Fe)@ZIF-8-MnOx-DOX and hM@ZMDF, hMIL-88B(Fe)@ZIF-8-MnOx-DOX-FAH. Concentration-dependent viability of MCF-7 cells incubated with (**C**) two pure MOFs, (**E**) hM@ZMD, and (**G**) hM@ZMDF. Concentration-dependent viability of HepG-2 cells incubated with (**D**) two pure MOFs, (**F**) hM@ZMD, and (**H**) hM@ZMDF. Reproduced with permission from [129]. Copyright (2021): American Chemical Society (Washington, DC, USA).

**Figure 11 biomedicines-12-00326-f011:**
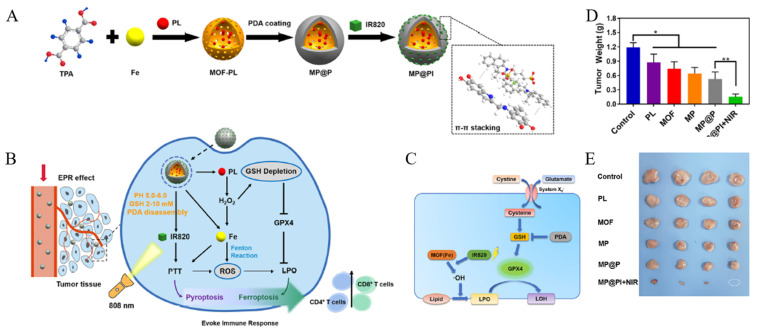
(**A**) Schematic preparation and antitumor mechanism of MP@Pi (**A**) Preparation of MP@PI nanoparticles. (**B**) Antitumor mechanism of MP@PI in vivo. (**C**) Specific mechanism of ferroptosis induced by MP@PI. (**D**) Weight and (**E**) representative photograph of dissected tumors. Reproduced with permission from (* *p* < 0.05, ** *p* < 0.01) [130]. Copyright (2022): American Chemical Society (Washington, DC, USA).

**Figure 12 biomedicines-12-00326-f012:**
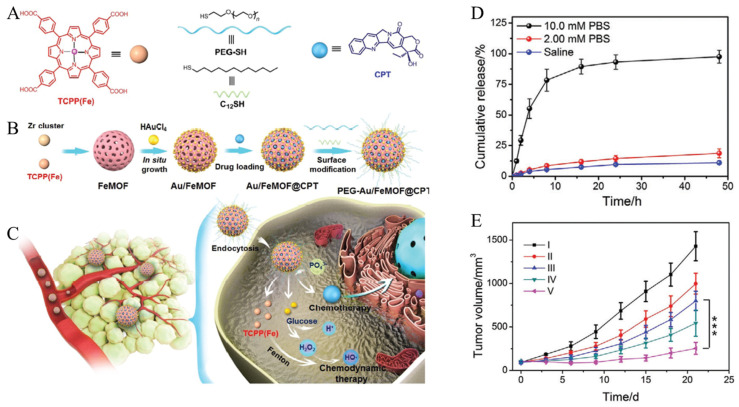
(**A**) Chemical structures of the building blocks (TCPP(Fe), PEG-SH, C_12_-SH, and CPT). (**B**) Preparation of the hybrid nanomedicine PEG-Au/FeMOF@CPT. (**C**) High tumor accumulation of PEG-Au/FeMOF@CPT NPs via passive targeting and subsequent cancer cell uptake. Triggered by intracellular phosphate, the chemotherapeutic drug CPT is released and a catalytic cascade of reactions is initiated. H_2_O_2_ generated through the oxidation of glucose by GNPs acts as chemical fuel for the Fenton reaction to produce highly toxic ROS and release the combination therapy. (**D**) Cumulative release curves of CPT from PEG-Au/FeMOF@CPT NPs in solutions containing different concentrations of phosphate ions. (**E**) Tumor growth curves of mice bearing HepG2 tumors who received different treatments. I, PBS; II, PEG-Au/FeMOF NPs; III, CPT; IV, PEG-Au/HMOF@CPT NPs; V, PEG-Au/FeMOF@CPT NPs (*** *p* < 0.001). Reproduced with permission from [131]. Copyright (2020): John Wiley and Sons (Hoboken, NJ, USA).

**Table 1 biomedicines-12-00326-t001:** Summary of the research on combination therapy using mesoporous nanoparticles.

Nanoparticle	Therapy	Functionalized Material	Functionalized Method	Drug	Ref.
HMSN	CHT PDT PTT	GPTMS CS FA	Siloxy bond Amide bond Epoxy–amine reaction	Dox PA	[103]
MSN	CHT PTT	MPTMS 3-mercaptopropionic acid PDA	Siloxy bonds Disulfide bond Self-polymerization	Dox	[105]
MSN	PDT PTT	PDA FA-PEG-SH	Self-polymerization Michael addition π−π stacking	ICG	[106]
XL-MSN	PTT IMT	APTMS GNP PEG-SH	Siloxy bonds Electrostatic interaction Au–thiol bond	CpG-ODN	[107]
HMSN	CHT PDT	APTMS HA	Siloxy bonds Schiff base bonds	Dox Rose bengal	[159]
HMSN	PTT RT	APTMS GNP HA-Dopamine	Siloxy bonds Electrostatic interaction Au–catechol bonds Amide bond	MnOx	[160]
PDA@MS	PTT IMT	1-tetradecanol	Phase change	Gardi	[113]
AgNP@MS	CHT CST	GOx	Amide bond	TPZ	[114]
GNR@MS	PTT IMT	VVP (97-mer peptide)	Amide bond	BMS1166	[115]
GNR@MS	PTT TDT	PEG Lauric acid	Physical adsorption. Phase change	AIPH	[116]
INP@MS	CHT MTT	CS FA	Disulfide bond Amide bond	Dox	[161]
CDs@MS	CHT PTT	Polyethyleneimine Trastuzumab	Amide bond	gemcitabine	[162]
PCN-224	CHT PDT	HA	Coordination bond	Dox	[128]
hMIL-88B(Fe)@ZIF-8	CHT CDT	FA MnOx	Coordination bond	Dox	[129]
MOF-235	PTT CDT	PDA IR820	Self-polymerization π−π stacking Hydrophobic interaction	PL	[130]
PCN-224 (Fe)	CHT CDT	PEG-SH C_12_-SH	Au–thiol bond π−π stacking, Coordination bond	CPT	[131]
mesoPt	CHT PTT	PEG	Pt–thiol bond	Dox	[136]
CDs	PTT PDT	TPP	Amide bond Electro static interaction	ALA	[145]
COF-366	PTT PDT	TAPP Terephthaldehyde	Imine bond	N/A	[153]
MPPD	PTT PDT	FA-PEG-SH	Michael addition	IR820 PFO	[158]
mesoPt	CHT PTT	Adamantane β-cyclodextrin	Pt–thiol bond Host–guest interaction	Dox	[163]
MPPD	CHT PTT	PEG-NH_2_	Michael addition Schiff base reactions	Dox	[164]

HMSN, hollow mesoporous silica nanoparticles; CHT, chemotherapy; PDT, photodynamic therapy; PTT, photothermal therapy; CS, chitosan; FA, folic acid; Dox, doxorubicin; PA, pheophorbide a; MSN, mesoporous silica nanoparticles; PDA, polydopamine; ICG, indocyanine green; XL-MSN, extra-large pore mesoporous silica nanoparticles; IMT, immunotherapy; GNP, gold nanoparticle; CpG-ODN, oligodeoxynucleotides containing unmethylated cytosine phosphorothioate-guanine motifs; HA, hyaluronic acid; RT, radiotherapy; MnOx, manganese oxide nanoparticles; Gardi, gardiquimod; AgNP, silver nanoparticle; CST, cancer starvation therapy; GOx, glucose oxidase; TPZ, tirapazamine; GNR, gold nanorod; TDT, thermodynamic therapy PEG, poly ethylene glycol; AIPH, 2,2′-azobis [2-(2-imidazolin-2-yl)propane]-dihydrochloride; INP, iron nanoparticle; MTT, magnetothermal therapy; CDs, carbon dots; CDT, chemodynamic therapy; PL, piperlongumine; CPT, camptothecin; mesoPt, mesoporous platinum nanoparticles; TPP, triphenylphosphonium; ALA, 5-aminolevulinic acid; COF, covalent organic frameworks; TAPP, tetra (p-amino-phenyl) porphyrin; MPPD, mesoporous polydopamine; PFO, perfluorooctane.

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
