# Peer review of "Advancing Cancer Treatment: Enhanced Combination Therapy through Functionalized Porous Nanoparticles"

_biomedicines, 2024, doi:10.3390/biomedicines12020326_

Round 1

Reviewer 1 Report

Comments and Suggestions for Authors

The manuscript introduces the ongoing challenge of cancer and the need for innovative treatment strategies, then focuses on the promising approach of using functionalized porous nanoparticles for combination therapy. Key details are provided on how combination therapy can enhance efficacy and mitigate the limitations of individual treatments. The unique properties of various porous nanoparticles are also highlighted along with how recent advances in functionalization can aid targeted delivery, controlled release, and overall efficacy of combination therapies. Recent advances in functionalization are also touched upon regarding how ligands, biomaterials, and polymers can tailor nanoparticles for effective combination therapy.

Overall, the manuscript orients the reader well to the purpose and focus of the review article. Relevant background and motivation set the stage for diving deeper into the synthesis, functionalization methods, and therapeutic applications of these porous nanoparticles across the remaining sections.

However, the writing style and organization for aided comprehension and information flow should be minorly checked by native speakers.

Moreover, expanding briefly on the specific limitations of current single-treatment methods may further strengthen the manuscript. But otherwise, all sections present the material clearly for the intended audience. Therefore, I would recommend this manuscript for publication after minor corrections.

Author Response

Comments to the Author
The manuscript introduces the ongoing challenge of cancer and the need for innovative treatment strategies, then focuses on the promising approach of using functionalized porous nanoparticles for combination therapy. Key details are provided on how combination therapy can enhance efficacy and mitigate the limitations of individual treatments. The unique properties of various porous nanoparticles are also highlighted along with how recent advances in functionalization can aid targeted delivery, controlled release, and overall efficacy of combination therapies. Recent advances in functionalization are also touched upon regarding how ligands, biomaterials, and polymers can tailor nanoparticles for effective combination therapy.

Overall, the manuscript orients the reader well to the purpose and focus of the review article. Relevant background and motivation set the stage for diving deeper into the synthesis, functionalization methods, and therapeutic applications of these porous nanoparticles across the remaining sections.

However, the writing style and organization for aided comprehension and information flow should be minorly checked by native speakers.

Moreover, expanding briefly on the specific limitations of current single-treatment methods may further strengthen the manuscript. But otherwise, all sections present the material clearly for the intended audience. Therefore, I would recommend this manuscript for publication after minor corrections.

Comment:

1) The writing style and organization for aided comprehension and information flow should be minorly checked by native speakers.

Answer: Thank you for your valuable comment. This manuscript was edited through an english editing service before submission. The certification was attached to the final page.

2) Expanding briefly on the specific limitations of current single-treatment methods may further strengthen the manuscript.

Answer: Thank you for the valuable comment. Limitation of single-treatment methods was added to manuscript as follow.

“For example, CHT induced multidrug resistance (MDR) which is one of the major causes of cancer treatment failure. CDT has the challenge of limited H2O2 content in the tumor microenvironment (TME). PDT has the challenge of low photosensitizer stability and high dependence on O2. PTT has limitations in treating cancer in deep tissues because light has difficulty penetrating deeply into tissues. The immune system activated by IMT can attack normal tissues, causing immune-related complications.” (Line 35-41)

Reviewer 2 Report

Comments and Suggestions for Authors

Since only one reference is from the year 2023,  I presume the years 2018 to 2022 are included. Although there are 161 references, some works that I am aware of have not been included (for example, Bing Xia et al. from Nanjing Forestry University, China).  A combination therapy is any therapy in which two or more treatments are applied. With respect to mesoporous silica nanoparticles, the key step in the possibility of their post-functionalization. Therefore I think that a review on functionalization of porous nanoparticles can be useful to a number of researchers. However, as written in Conclusion, " ... Various methods utilizing MSN, NP@MSN, MOF, mesoPt, CDs, COF, and MPPD 545 have been explored to facilitate combination therapy. ... ", not only MNP are discussed and therefore it might be wise to change the title of the review emphasizing the concept of combination therapy. 

Table 1 is important part of the review because it offers a quick and concise glance of the the subject reviewed, but to be fully useful it should be accompanied with the explanation of all acronyms, no matter they have already been defined in the main text.

Author Response

Comments to the Author
Since only one reference is from the year 2023, I presume the years 2018 to 2022 are included. Although there are 161 references, some works that I am aware of have not been included (for example, Bing Xia et al. from Nanjing Forestry University, China).  A combination therapy is any therapy in which two or more treatments are applied. With respect to mesoporous silica nanoparticles, the key step in the possibility of their post-functionalization. Therefore, I think that a review on functionalization of porous nanoparticles can be useful to a number of researchers.

However, as written in Conclusion, " ... Various methods utilizing MSN, NP@MSN, MOF, mesoPt, CDs, COF, and MPPD 545 have been explored to facilitate combination therapy. ... ", not only MNP are discussed and therefore it might be wise to change the title of the review emphasizing the concept of combination therapy.

Table 1 is important part of the review because it offers a quick and concise glance of the the subject reviewed, but to be fully useful it should be accompanied with the explanation of all acronyms, no matter they have already been defined in the main text.

Comment:

1) Since only one reference is from the year 2023, I presume the years 2018 to 2022 are included. Although there are 161 references, some works that I am aware of have not been included (for example, Bing Xia et al. from Nanjing Forestry University, China).  A combination therapy is any therapy in which two or more treatments are applied. With respect to mesoporous silica nanoparticles, the key step in the possibility of their post-functionalization. Therefore, I think that a review on functionalization of porous nanoparticles can be useful to a number of researchers.

Answer: Thank you for the valuable comment. Reference paper (ACS Nano 2023,17,1036-1053.) was added as follow.

“Therefore, research efforts have also been directed towards overcoming these issues by uti-lizing mesoporous nanoparticles [79-84].” (Line 79-84)

  1. Li, J.; Fan, J.; Gao, Y.; Huang, S.; Huang, D.; Li, J.; Wang, X.; Santos, H.A.; Shen, P. ; Xia, B.,Porous Silicon Nanocarriers Boost the Immunomodulation of Mitochondria-Targeted Bovine Serum Albumins on Macrophage Polarization. ACS Nano 2023,17,1036-1053. (Line 770-772)

2) It might be wise to change the title of the review emphasizing the concept of combination therapy.

Answer: Thank you for your valuable comment. Title was changed as follow.

“Advancing cancer treatment: Enhanced combination therapy through functionalized porous nanoparticles” (Line 2-3)

3) Table 1 is important part of the review because it offers a quick and concise glance of the the subject reviewed, but to be fully useful it should be accompanied with the explanation of all acronyms, no matter they have already been defined in the main text.

Answer: Thank you for the valuable comment. Explanation of all acronyms was added as follow.

Table 1. Summarization of research on combination therapy using mesoporous nanoparticles.

Nanoparticle

Therapy

Functionalized material

Functionalized method

Drug

Ref

HMSN

CHT

PDT

PTT

GPTMS

CS

FA

Siloxy bond

Amide bond

Epoxy-amine reaction

Dox

PA

[103]

MSN

CHT

PTT

MPTMS

3-mercaptopropionic acid

PDA

Siloxy bonds

Disulfide bond

Self-polymerization

Dox

[105]

MSN

PDT

PTT

PDA

FA-PEG-SH

Self-polymerization

Michael addition

π−π stacking

ICG

[106]

XL-MSN

PTT

IMT

APTMS

GNP

PEG-SH

Siloxy bonds

Electrostatic interaction

Au-thiol bond

CpG-ODN

[107]

HMSN

CHT

PDT

APTMS

HA

Siloxy bonds

Schiff base bonds

Dox

Rose bengal

[159]

HMSN

PTT

RT

APTMS

GNP

HA-Dopamine

Siloxy bonds

Electrostatic interaction

Au–catechol bonds

Amide bond

MnOx

[160]

PDA@MS

PTT

IMT

1-Tetradecanol

Phase change

Gardi

[113]

AgNP@MS

CHT

CST

GOx

Amide bond

TPZ

[114]

GNR@MS

PTT

IMT

VVP (97-mer peptide)

Amide bond

BMS1166

[115]

GNR@MS

PTT

TDT

PEG

Lauric acid

Physical adsorption.

Phase change

AIPH

[116]

INP@MS

CHT

MTT

CS

FA

Disulfide bond

Amide bond

Dox

[161]

CDs@MS

CHT

PTT

Polyethyleneimine

Trastuzumab

Amide bond

gemcitabine

[162]

PCN-224

CHT

PDT

HA

Coordination bond

Dox

[128]

hMIL-88B(Fe)@ZIF-8

CHT

CDT

FA

MnOx

Coordination bond

Dox

[129]

MOF-235

PTT

CDT

PDA

IR820

Self-polymerization

π−π stacking

hydrophobic interaction

PL

[130]

PCN-224 (Fe)

CHT

CDT

PEG-SH

C12-SH

Au-thiol bond

π−π stacking,

Coordination bond

CPT

[131]

mesoPt

CHT

PTT

PEG

Pt-thiol bond

Dox

[136]

CDs

PTT

PDT

TPP

Amide bond

Electro static interaction

ALA

[145]

COF-366

PTT

PDT

TAPP

terephthaldehyde

Imine bond

N/A

[153]

MPPD

PTT

PDT

FA-PEG-SH

Michael addition

IR820

PFO

[158]

mesoPt

CHT

PTT

Adamantane

β-cyclodextrin

Pt-thiol bond

Host-Guest interaction

Dox

[163]

MPPD

CHT

PTT

PEG-NH2

Michael addition

Schiff base reactions

Dox

[164]

(HMSN, hollow mesoporous silica nanoparticles; CHT, chemotherapy; PDT, photodynamic therapy; PTT, photothermal therapy; CS, chitosan; FA, folic acid; Dox, doxorubicin; PA, pheophorbide a; MSN, Mesoporous silica nanoparticles; PDA, polydopamine; ICG, indocyanine green; XL-MSN, extra-large pore mesoporous silica nanoparticles; IMT, immunotherapy; GNP, gold nanoparticle; CpG-ODN, oligodeoxynucleotides containing unmethylated cytosine phosphorothioate-guanine motifs; HA, hyaluronic acid; RT, radiotherapy; MnOx, manganese oxide nanoparticles; Gardi, gardiquimod; AgNP, silver nanoparticle; CST, cancer starvation therapy; GOx, glucose oxidase; TPZ, tirapazamine; GNR, gold nanorod; TDT, thermodynamic therapy PEG, poly ethylene glycol; AIPH, 2,2′-azobis[2-(2-imidazolin-2-yl)propane]-dihydrochloride; INP, iron nanoparticle; MTT, magnetothermal therapy; CDs, carbon dots; CDT, chemodynamic therapy; PL, piperlongumine; CPT, camptothecin; mesoPt, mesoporous platinum nanoparticles; TPP, triphenylphosphonium; ALA, 5-aminolevulinic acid; COF, covalent organic frameworks; TAPP, tetra (p-amino-phenyl) porphyrin; MPPD, mesoporous polydopamine; PFO, perfluorooctane)

(Line 570-582)